# Molecular Changes in the Brain of the Wintering *Calidris pusilla* in the Mangroves of the Amazon River Estuary

**DOI:** 10.3390/ijms241612712

**Published:** 2023-08-12

**Authors:** Patrick Douglas Corrêa Pereira, Ediely Pereira Henrique, Emanuel Ramos da Costa, Anderson de Jesus Falcão, Mauro André Damasceno de Melo, Maria Paula Cruz Schneider, Rommel Mario Rodriguez Burbano, Daniel Guerreiro Diniz, Nara Gyzely de Morais Magalhães, David Francis Sherry, Cristovam Wanderley Picanço Diniz, Cristovam Guerreiro-Diniz

**Affiliations:** 1Laboratório de Biologia Molecular e Neuroecologia, Campus Bragança, Instituto Federal de Educação, Ciência e Tecnologia do Pará, Bragança 68600-000, PA, Brazil; pereira.d.c.patrick@gmail.com (P.D.C.P.);; 2Laboratório de Citogenética Humana, Universidade Federal do Pará, Belém 66075-110, PA, Brazil; 3Laboratório de Investigações em Neurodegeneração e Infecção, Hospital Universitário João de Barros Barreto, Instituto de Ciências Biológicas, Universidade Federal do Pará, Belém 66075-110, PA, Brazil; 4Laboratório de Microscopia Eletrônica, Seção de Hepatologia, Instituto Evandro Chagas, Belém 66093-020, PA, Brazil; 5Department of Psychology, Advanced Facility for Avian Research, University of Western Ontario, London, ON N6G 1G9, Canada

**Keywords:** brain transcriptome, migratory birds, spring migration, wintering, *Calidris pusilla*

## Abstract

Migrant birds prepare differently to fly north for breeding in the spring and for the flight to lower latitudes during autumn, avoiding the cold and food shortages of the Northern Hemisphere’s harsh winter. The molecular events associated with these fundamental stages in the life history of migrants include the differential gene expression in different tissues. Semipalmated sandpipers (*Calidris pusilla*) are Arctic-breeding shorebirds that migrate to the coast of South America during the non-breeding season. In a previous study, we demonstrated that between the beginning and the end of the wintering period, substantial glial changes and neurogenesis occur in the brain of *C. pusilla*. These changes follow the epic journey of the autumn migration when a 5-day non-stop transatlantic flight towards the coast of South America and the subsequent preparation for the long-distance flight of the spring migration takes place. Here, we tested the hypothesis that the differential gene expressions observed in the brains of individuals captured in the autumn and spring windows are consistent with the previously described cellular changes. We searched for differential gene expressions in the brain of the semipalmated sandpiper, of recently arrived birds (RA) from the autumnal migration, and that of individuals in the premigratory period (PM) in the spring. All individuals were collected in the tropical coastal of northern Brazil in the mangrove region of the Amazon River estuary. We generated a de novo neurotranscriptome for *C. pusilla* individuals and compared the gene expressions across libraries. To that end, we mapped an RNA-Seq that reads to the *C. pusilla* neurotranscriptome in four brain samples of each group and found that the differential gene expressions in newly arrived and premigratory birds were related with neurogenesis, metabolic pathways (ketone body biosynthetic and the catabolic and lipid biosynthetic processes), and glial changes (astrocyte-dopaminergic neuron signaling, astrocyte differentiation, astrocyte cell migration, and astrocyte activation involved in immune response), as well as genes related to the immune response to virus infections (Type I Interferons), inflammatory cytokines (IL-6, IL-1β, TNF, and NF-κB), NLRP3 inflammasome, anti-inflammatory cytokines (IL-10), and cell death pathways (pyroptosis- and caspase-related changes).

## 1. Introduction

Each year, migratory birds undergo behavioral, cellular, and molecular changes associated with migration and reproduction [1,2,3,4,5]. During autumn, they fly towards lower latitudes for the wintering period, where they find milder temperatures and food, and experience progressive functional changes at all levels in preparation for the spring migration, when they fly north for reproduction [6,7,8]. The spring migration requires a preparation for the long-distance migratory flights by fueling at the wintering sites, when fat accumulation, metabolic enzymatic changes, and lipogenesis in the liver with subsequent transport to the skeletal muscle indicates a readiness for departure [9]. 

Previous reports in different species investigated migratory changes at the molecular [10,11], cellular [12,13,14,15], and systemic changes [16,17,18,19]. These adaptive responses take hold during spring and autumn migrations, when migratory birds shift between two life history states (LHS) [20], with contrasting seasonal phenotypic profiles that emerge before and after reproduction, respectively [6]. These states occur in association with distinct regulatory strategies at the transcriptional level in autumn and spring based on the differential expressions of hypothalamic genes. [4,21]. For example, after seasonal LHS photoperiod induction, the Black-headed Bunting songbird, *Emberiza melanocephala*, showed significant differences between the spring premigratory and postmigratory phenotypes in the activity-rest pattern, body fattening, weight gain, testis size, heart and intestine weights, blood glucose, and triglyceride levels [6]. 

Many species of shorebirds migrate to low-latitude climate zones around the equator to escape the harsh northern winter climate and lack of food. This tropical region around the equator, which remains stable from year to year, is an area rich in food resources and has warmer temperatures with little annual variation, allowing migratory birds to recover from the autumnal long journey and prepare for the long migratory flights in spring and [22,23,24,25]. 

Due to the stability of the environment at lower latitudes, the endogenous rhythms imposed by the internal clocks in combination with the wintering and stopover locations weather, wind favorable conditions and temperature raise may determine the timing of the vernal migration [26,27,28,29,30,31]. 

The semipalmated sandpiper *C. pusilla* performs a remarkable five to six days nonstop flight across the Atlantic Ocean from James Bay (Ontario, Canada) or from the Bay of Fundy (between New Brunswick and Nova Scotia, Canada) to coastal South America, Caribbean, and Central America, before moving on to their wintering area in Brazil [25,32]. This species arrives on the coast of Venezuela [33] and in the Brazilian Coast [34] between the middle of August and early September and stays at resource-rich locations until April/May, when birds start the vernal migration [33]. In these areas, these migratory shorebirds spend a large portion of the non-breeding season [35], where they exchange feathers, increase body mass [36], and decrease corticosterone levels [33], maximizing their fitness in preparation for the next long flight of spring migration. 

There is not a single report related to brain molecular changes associated with the wintering period following the long, uninterrupted migratory flight across the Atlantic Ocean (in fall), or the preparation for the multiple stopover migratory flight for breeding (in spring) in this species. Similarly, the brain transcriptome before and after reproduction remains to be investigated. Since the ribonucleic acids represent the genomic expressions by linking the genotype to the phenotype [37], we compared two snapshots of transcripts in the brains of recently arrived and pre-migratory semipalmated sandpiper (*Calidris pusilla*), captured respectively at two time windows of the wintering period: August/September (fall) and April/May (spring). 

We have previously demonstrated that the autumn migration and subsequent recovery to the spring migration promotes substantial glial changes and neurogenesis in the brain of *C. pusilla* [12,14,38]. Here, we tested the hypothesis that the differential gene expressions observed in the brains of individuals captured in the autumn and spring windows during the wintering period at the mangrove region of the estuary of the Amazon River are in line with those changes.

In the absence of a sequenced genome to guide the reconstruction process, the transcriptome was assembled de novo based on RNA-sequencing reads (RNA-Seq) and annotation [39]. We searched for differential gene expressions in the brain of this latitudinal migrant species, and the results were used to interpret the functional implications of the genomic expressions [40].

## 2. Results

### 2.1. Sequencing Assembly of Semipalmated Sandpiper Transcriptome

The RNA telencephalon tissues of *C. pusilla* were sequenced in an Ion Proton Sequencer. Eight samples generated a total of 130,275,514 single-end raw reads, of these, 66.1% survived the trimming/short-reads removal phase and were used for transcriptome assembly, producing a total of 266,414 transcripts. 

### 2.2. Gene Expression between Experimental Groups

The transcript expression data were obtained via mapping back the reads to the assembled transcripts. The volcano plot in Figure 1 exhibits the statistical significance of the difference relative to the amplitude of difference for every single gene in the comparison between recently arrived and premigratory birds’ brains, through the negative base-10 log and base-2 log fold-change, respectively. See [41] for a detailed explanation of the volcano plot representation. The present report revealed 615 differentially expressed genes (DEGs) in the brains of recently arrived and premigratory groups: 356 upregulated and 259 downregulated genes (Figure 1 and Appendix A).

Figure 2 is a large-scale snapshot of the genomic differential expressions in the brains of RA and PM *C. pusilla* datasets. The heat map of the expression shows two clusters of genes with opposed patterns, showing a unique profile for each cluster among the experimental groups (Figure 2 and Appendix A shows an expanded version of the heatmap, as presented in Figure 2).

### 2.3. Gene Ontology and Functional Analysis

A gene ontology (GO) annotation analysis was performed for the brain transcriptome of the RA and PM *C. pusilla* (the full annotation report is in Appendix A). We found a total of 4656 enriched terms for the RA birds and 1859 terms for the PM birds. From these numbers, 923 belongs to Cellular Component (CC); 1730 to Molecular Function (MF); and 3862 to Biological Process (BP). The terms annotated in the gene ontology (GO) for the transcriptome showed different functional roles that may reflect pre- and post-breeding fundamental stages in the life history of this long-distance migrant species. Table 1 lists the top 11 differentially expressed genes with their respective enriched GO terms related to the biological process exhibiting significant differential expressions.

Previous analysis of the genes illustrated in Table 1 in other species revealed their involvement in a variety of biological processes that, when examined from the perspective of differential gene expressions in the brain of *C. pusilla*, raises relevant questions about the contribution of the wintering period for the spring migration. Indeed, the top six upregulated (*GABBR2*, *MAN1A2*, *BCAT1*, *NXPE3*, *FGF9*, and *TRDMT1*) and the top four downregulated (*PLXNA1*, *BICRAL*, *ARHGEF9*, and *MVB12B*) differentially expressed genes in the brains of RA and PM individuals are involved in distinct biological processes, as indicated by GO names in Table 1. 

In addition, another group of genes (not included in Table 1) related to the phenotypic metabolic changes of migrant birds were also found as differentially expressed: the ZNF703 gene, associated with ketone body biosynthetic and catabolic processes upregulated in RA individuals; AGMO, associated with the lipid biosynthetic process upregulated in PM birds. Upregulated genes in RA birds were also related with astrocyte-dopaminergic neuron signaling (ATXN1), astrocyte cell migration (SCRIB), and astrocyte activation and differentiation (ZNF703). In contrast, in PM birds, upregulated genes related to astrocytes functions were limited to the gene associated with the activation for immune response (APP) and the gene CTNNB1 associated with astrocyte-dopaminergic neuron signaling. 

Finally, we found a positive regulation of genes related to immune response to virus infections (Type I Interferons, inflammatory cytokines (IL-6, TNF and NF-κB), NLRP3 inflammasome, anti-inflammatory cytokines (IL-10), and cell death pathways (pyroptosis and apoptosis)) in RA individuals.

## 3. Discussion

In this study, we performed de novo assembly of RNA extracted from the brains of *C. pusilla* that were collected before and after breeding, seven months apart, during the wintering period, and we compared the transcriptomic changes. Samples were collected in September/October, when the birds had just completed their autumnal migration, and in April/May, when the birds became ready for the spring migration. We found 259 upregulated and 357 downregulated genes differentially expressed in the brains of recently arrived and pre-migratory birds. We confirmed the hypothesis that the brain molecular changes observed in recently arrived and premigratory birds during the wintering period were coherent with previously described cellular changes in the same species and time windows [12,14,38]. In addition, significant changes were found in the differential expressions of genes related to the inflammatory and anti-inflammatory response to virus infections, and this included Type I Interferons, IL-6, IL-1β, TNF, and NF-κB, NLRP3 inflammasome, anti-inflammatory cytokines (IL-10) and cell death pathways (pyroptosis- and caspase-related apoptosis).

### 3.1. GABBR2 and ARHGEF9 Differential Expressed Genes in Wintering C. pusilla

Notably, *GABBR2* and *ARHGEF9* exhibit contrasting differential gene expressions related to the inhibitory activity in the brain of *C. pusilla* (Figure 1). The upregulation of the synthesis of Subunit 2 of the Gamma-Aminobutyric Acid Receptor Type B may provide an increase in the GABAergic transmission in the brain as the spring migration approaches. It is important to highlight that the hippocampal circuits involved in the learning and memory [43] and social interaction [44] seem to be important for the long flights of migrations. Social interaction between birds of the same group may facilitate collective behavior to form flocks and organize flights during the migration for energy savings [45]. Of note, in a previous study, we found in a shorebird of the same family (Scolopacidae) a significant increase in parvalbuminergic neurons in the hippocampal formation of this species, in the premigratory groups captured in the wintering period, at the same time point and place [46]. If this observation about the PV neurons of the hippocampal formation extends to *C. pusilla*, it is reasonable to raise the hypothesis that GABAergic adaptive changes may be required for the spring migratory flight. In contrast, since autumn migration has been left behind, and long-distance flights will not be required during the wintering period, GABAB expressions may not be required to the same extent. In addition, parvalbumin-expressing basket-cell network plasticity induced by experience regulates adult learning [47], and hippocampal early-born and late-born PV neurons are recruited in rule consolidation and new information acquisition through the excitation and inhibition, respectively [48,49]. Learning and memory is modulated via hippocampal GABAergic activity through the GABB receptor and metabotropic glutamate receptor-dependent cooperative long-term potentiation, suggesting that GABAergic synapses may contribute to functional synaptic plasticity in the adult hippocampus [50,51]. Indeed, the synaptic plasticity of inhibitory neurons provides long-lasting changes in the hippocampal network, and this is a key component of memory formation [52,53]. Distinct interneuron types contribute to the temporal binding of hippocampal ensembles, synaptic plasticity, and the acquisition of spatial and contextual information [53,54]. This hippocampal activity is part of the neuronal network used to integrate the multisensory navigational information (magnetic field, celestial cues, and geographical cues) in the hippocampal formation, and this is important to define flight direction and stopover recognition during the spring long-distance migration, back to the breeding niches [55,56,57]. 

In line with these findings is the differential downregulation of the Guanine Nucleotide Exchange Factor *ARHGEF9* that is essential for the synaptic localization and maintenance of GABAA receptors in the postsynaptic neuronal membrane in the hippocampus [58,59,60]. This gene synthesizes collybistin, a guanine nucleotide exchange factor that seems essential to materialize this operation [58]. Important to remember is that GABA type A receptor is a ionotropic ligand-gated chloride channel which mediates fast inhibitory signals through rapid postsynaptic hyperpolarization, whereas GABA type B is a metabotropic receptor producing slow and prolonged inhibitory signals via G proteins and second messengers involved in pre- and postsynaptic inhibition, the regulation of Ca^2+^, and K channels [61]. 

Thus, related to brain inhibition in this species, many questions emerge to be explored in future studies. For example: what is the functional role of such contrasting differential gene expressions for the GABAA and GABAB receptors found in the brains of newly arrived and pre-migratory individuals? It is important to highlight that although our samples were separated by 7 to 8 months (wintering period), the time windows studied were close to the reproductive period. In fact, the autumn migration takes place two months after breeding (August and September), and the spring migration back to the breeding site takes place two months before breeding (April and May). Due to this proximity to the reproduction period, we can ask whether the physiological changes induced before and after reproduction would differentially modulate these receptors in the brain. As the expression of the GABA_A_ receptor subunit transcriptional regulation is affected by sexual hormones [62], and as hormone levels may be not by the same before and after reproduction [63], the differential expressions of the GABAA receptors may change. It remains, however, to be investigated in detail, the physiological implications of these differential gene expressions regulating the GABAergic receptors in the wintering *C. pusilla* at lower latitudes.

Since our sample did not distinguish the different areas of the nervous system, nor did it examine the expression in different neuronal types, it is worth asking whether these receptors are differentially expressed in different brain areas and in different neuronal types in further investigations.

### 3.2. Gene Differential Expressions Related with Metabolic Pathways, Glial Changes, Neurogenesis, and Anti-Virus Response in Recently Arrived and Premigratory C. pusilla

Migratory birds have no access to supplementary water or nutrition during non-stop multi-day flight, and body fat and protein stores provide both fuel and life support. Long-distance migratory flights require lipid reserves [64], and they must have everything on board before departure [65]. Indeed, it has been demonstrated that higher levels βOHB in the blood, as is expected during the fasting period, is associated with the transatlantic flight, can meet all basal requirements, and only around half of the energy is necessary for neuronal activity [66]. *Calidris pusilla* must fast for 5–6 days during the long flight [32], and when glucose is in short supply, the brain increases ketone body metabolism [66]. The brains of migratory birds can adapt to the utilization of ketone bodies for its energy requirements during the fasting state, and this may lead to a high demand for astrocytes to take up, synthesize, and release fatty acids, which are alternative sources of energy that can be released as β-hydroxybutyrate, a ketone body that can fuel brain cells, including astrocytes, neurons, and oligodendrocytes [67]. Coherently, we found in the present report differential gene expressions related to ketone body biosynthetic and catabolic processes (ZNF703) in recently arrived individuals, whereas differential expressions of the lipid biosynthetic process (AGMO) were found in premigratory birds. 

Accordingly, in previous reports, we hypothesized that long flights may affect astrocytes, and we previously demonstrated the contrasting astrocyte changes in recently arrived and premigratory *C. pusilla* [14]. In the present report, we found that astrocyte-related genes were differentially expressed, indicating the upregulation of astrocyte-dopaminergic neuron signaling (ATXN1), astrocyte cell migration (SCRIB), and astrocyte activation and differentiation (ZNF703) in recently arrived individuals, whereas differential gene expressions in premigratory birds related to astrocytes were limited to astrocyte activation involved in immune response (APP) and astrocyte-dopaminergic neuron signaling (CTNNB1). We have no explanation for the contrasting expressions of CTNNB1 and ATXN1. 

We have recently searched for virus transcripts in the brain of *C. pusilla* using the pipeline VIRTUS (v.1.2.1) and VIRTUS2 (v.2.0), using as reference the genome and transcriptome of *Calidris pugnax* (Accession code ASM143184v1), followed by NCBI Ref-Seq Viral Genomes [68]. We found 370 virus species in the brain of *C. pusilla*, three of which (Simbu orthobunyavirus—NC_018477, *Choristoneura fumiferana* granulovirus—NC_008165, and Shamonda orthobunyavirus—NC_018464) were differentially expressed [69]. The stopovers in bird migration may contribute to the recovery of the constitutive immune function, which is compromised during migration [70], whereas a non-stop transatlantic flight may not allow the innate immune system to recover before the arrival at the final destination and may affect the ability of the bird, in this instance, to clear a virus [17]. 

As the primary function of the immune system is, however, the recognition and elimination of the invading pathogens (resistance) [71,72], or alternatively, the control of the damage induced by a given burden (tolerance) [73], we suggest that *C. pusilla* may have developed these immunological mechanisms and may function as a virus reservoir. 

The interferon system provides the first line of defense against viral infection in vertebrates with type I IFN, promoting humoral immunity [74], and both type I and III IFNs, associated with the adoption of an anti-viral state in infected and neighboring cells [75,76]. Inflammatory cytokine production follows the signaling pathways activated by viral (mainly DNA and RNA virus) pathogen-associated molecular patterns [77]. 

In the present report, we found the positive regulation of NLRP3 inflammasome complex assembling in recently arrived *C. pusilla*, suggesting an ongoing inflammation. NLR family pyrin domain containing 3 (NRLP3) has been linked to viral-induced inflammation [78]. Inflammasomes activate inflammatory caspases promoting the maturation of IL-1β and IL-18, while inducing cell death by pyroptosis [79]. Though some inflammasome genes were also expressed in birds, little is known about the role of inflammasomes in avian responses [80]. 

Coherently, with differential virus transcripts in the brain of *C. pusilla*, we found significant differential expressions of genes related to the inflammatory and anti-inflammatory response to virus infections, and this included Type I Interferons, IL-6, IL-1β, TNF, and NF-κB, anti-inflammatory cytokines (IL-10), and cell death pathways (pyroptosis- and caspase-binding).

#### Concluding Remarks

In the present work, we used information from the genomic studies of other species to speculate about the functional significance of the genes of *C. pusilla* that are being differentially expressed through wintering in recently arrived (in the fall) and premigratory (spring) individuals. To our knowledge, this is the first study demonstrating differential gene expression patterns in the brain tissue of migratory birds in natural environments throughout two different time windows of the wintering period, leading to phenotypic changes.

Gene expressions and phenotypes seem to reflect the bird physiology in these two time points of the wintering period. Indeed, we found contrasting differential expressions of genes that regulate the inhibitory neuronal activity in the brain, phenotypic expressions of astrocytes, metabolic pathways, and innate immune response. 

Further studies dedicated to brain-specific gene expression signatures and environment-mediated changes, in correlation with recently arrived and premigratory bird’s gene expression patterns, may elucidate the complex network of modifications to the brain transcriptome. 

## 4. Materials and Methods

### 4.1. Bird Sampling and Ethics Recommendations for the Use of Animals in Research

We used a total of eight *C. pusilla* individuals, all collected in Otelina Island (0°45′42.57″ S and 46°55′51.86″ W), four of which were collected during the period between September and October (Recently arrived birds), and the other four between April and May (Premigratory birds). Birds were captured under license N° 44551-2 from the Chico Mendes Institute for the conservation of Biodiversity (ICMBio). All procedures were carried out in accordance with the Association for the Study of Animal Behavior/Animal Behavior Society Guidelines for the Use of Animals in Research. All efforts were made to minimize the number of animals used, stress, and discomfort.

### 4.2. Transcardiac Perfusion with RNA Later and RNA-Sequencing

The birds were transcardially perfused with a saline solution, followed by RNA Later^®^ Solution. The brains were removed from the skull and stored at −20 °C prior to sequencing. Telencephalic tissues were homogenized for extraction and sequencing. The total RNA was extracted according to the manufacturer’s suggested protocol for isolating the RNA from the tissues, and the mRNA was isolated and purified using the Dynabeads™ mRNA DIRECT™ Micro Purification kit (Thermo Fisher Scientific, São Paulo, SP, Brazil). For the conversion of RNA into cDNA, we used the Ion Total RNA-Seq Kit v2 (Thermo Fisher Scientific, São Paulo, SP, Brazil). The template preparation was performed in the Ion One Touch 2 Instrument with the OT2 200 kit (Life Technologies, São Paulo, SP, Brazil). The fragment sequencing was performed in the Ion PI chip v2 via the Ion Proton System Instrument. Eight single-ends read FASTQ files were generated that referenced the eight biological replicates (four individuals from the RA group and four from the PM group).

### 4.3. Filtering, Trimming, and Transcriptome Assembly

To verify the quality of the sequenced transcriptome, we used the software FastQC 0.18 [81], and for cleaning up the low-quality reads, we used the Trimmomatic 0.36 [82]. To perform the de novo assembly, we used the Trinity 2.11 [83] according to the default parameters and the Salmon v1.0 software to quantify the transcript expressions [84]. 

### 4.4. Differential Expression Discovery and Functional Annotation

We used the EdgeR v3.38.4 package to run the differential expression analysis [85] and Blast2GO [86] to identify the differentially expressed transcripts via several Blast+ v2.14.0 [87] searches. In an attempt to identify the maximum number of valid hits, we blasted our differentially expressed transcripts using Uniprot SwissProt and TrEMBL [88], NCBI non-redundant proteins and nucleotides (nr and nt) [89], RefSeq Protein and RNA [89], Uniref [90], and GenPept [91]. After the Blast+ phase, we ran the Blast2GO v6.0 functional annotation protocol according to the user manual and default parameters [86,92]. 

The volcano plot exhibits a false discovery rate (FDR) as their significance values (*y*-axis) [93], which were transformed to their −log10. The higher the position of a point, the more significant its value. The positive fold change values (to the right) correspond to the upregulated and negative fold change values (to the left), which represents the downregulated genes. The vertical line at log 2 ratio = zero indicates fold change = one. Thresholds were based on the following cutoffs: FDR-values below 0.01 and log fold-changes above 4. In this plot, each dot (regardless of color) represents a gene, and the color of each dot defines its significance in relation to the levels set in the graph. Gray dots represent genes that did not show a differential expression (not significant), and blue dots are significant only for Log2FC > 4. Red and green dots above the horizontal dashed line are significant for both Log2FC and FDR and highlights genes with FDR < 0.01 and fold changes above 4. The top 11 differentially expressed genes are indicated with their labels and their expression values are indicated as TMM (Trimmed Mean of M-values; M = log2RA/PM). 

## 5. Conclusions

In the present work, we compared the brain transcriptomes of *C. pusilla* at two time points of the winter period: after breeding and a long-distance uninterrupted autumnal flight across the Atlantic Ocean, and before breeding, at the end of the winter period and before the spring migratory flight back to the reproductive site [11,94]. We quantified differentially expressed genes of recently arrived and premigratory individuals and used information from the genomic studies of other species to speculate about the functional significance of this differential expression. The underlying assumption was that genomic functional regions are conserved in birds and mammals through evolution [95,96]. Although this is a limited way to interpret the results [96,97], it is reasonable to use this approach as a first step towards a detailed molecular-based guide for formulating and testing new hypotheses. We have previously demonstrated that the autumn migration and subsequent recovery to the spring migration promotes substantial glial changes and neurogenesis in the brain of *C. pusilla* [14,38]. Here, we confirmed the hypothesis that the differential gene expressions observed in the brains of individuals captured in the autumn and spring windows accompany phenotypic changes previously described for this species in the same wintering periods.

Since the functional genomic analysis referred to in the present report is based on different species, and the whole genome of *C. pusilla* is not available, a deeper understanding of the biology of its transcriptome and functional implications are necessary.

## Figures and Tables

**Figure 1 ijms-24-12712-f001:**
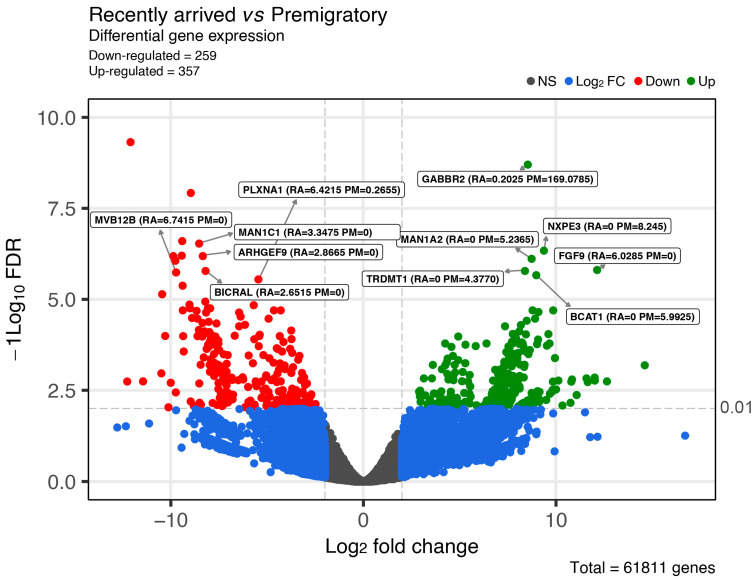
Volcano plot of differentially expressed genes between the pairwise comparison of premigratory and recently arrived groups. The vertical axis shows −log10 adjusted *p*-value (FDR logarithmized significant *p*-values) and the horizontal axis shows the Log_2_Fold Change (change in the gene expression level). Eleven of the greater top 20 differentially expressed genes are identified by indicating between brackets, normalized correspondent values between samples comparisons for recently arrived (RA) and premigratory (PM) groups. Abbreviations: *GABBR2* = Gamma-Aminobutyric Acid Type B Receptor Subunit 2; *NXPE3* = Neuroexophilin and PC-Esterase domain family member 3; *FGF9* = Fibroblast Growth Factor 9; *TRDMT1* = TRNA Aspartic Acid Methyltransferase 1; *BCAT1* = Branched Chain Amino Acid Transaminase 1; *ARHGEF9* = Cdc42 Guanine Nucleotide Exchange Factor 9; *MVB12B* = Multivesicular Body Subunit 12B; *BICRAL* = BICRA like Chromatin Remodeling Complex Associated Protein; *PLXNA1* = Plexin A1, *MAN1C1* = Mannosidase Alpha Class 1C Member 1; *MAN1A2* = Mannosidase Alpha Class 1A Member 2.

**Figure 2 ijms-24-12712-f002:**
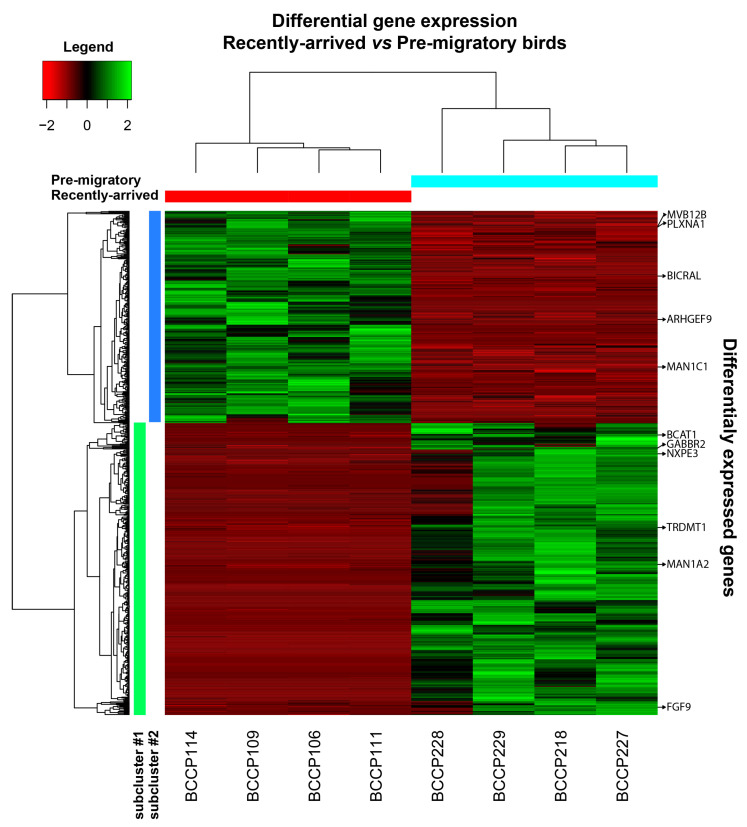
Hierarchical cluster analysis of the differential gene expressions (DEGs) and transcriptome RNA-sequencing heat maps of these genes in the brains of recently arrived (RC) and pre-migratory (PM) *Calidris pusilla* (adjusted p-cutoff of 0.01 for classification as differentially expressed). Top: Dendrogram of the distribution of individuals according to differentially expressed genes in the brains of RC (red bar) and PM (blue bar) (same dataset in Figure 1). The detailed dendrogram on the left identifies upregulated (green) and downregulated (red) genes in each biological replicate of distinct RC and PM individuals. The brighter the color, the more differentially expressed a gene is. Colored bars under the left dendrogram (light green and blue) indicate the two groups of genes with contrasting expression patterns (see Figure 3). Bottom: Individual identifications on subclusters 1 and 2. Abbreviations: *GABBR2* = Gamma-Aminobutyric Acid Type B Receptor Subunit 2; *NXPE3* = Neuroexophilin and PC-Esterase domain family member 3; *FGF9* = Fibroblast Growth Factor 9; *TRDMT1* = TRNA Aspartic Acid Methyltransferase 1; *BCAT1* = Branched Chain Amino Acid Transaminase 1; *ARHGEF9* = Cdc42 Guanine Nucleotide Exchange Factor 9; *MVB12B* = Multivesicular Body Subunit 12B; *BICRAL* = BICRA like Chromatin Remodeling Complex Associated Protein; *PLXNA1* = Plexin A1; *MAN1C1* = Mannosidase Alpha Class 1C Member 1; *MAN1A2* = Mannosidase Alpha Class 1A Member 2.

**Figure 3 ijms-24-12712-f003:**
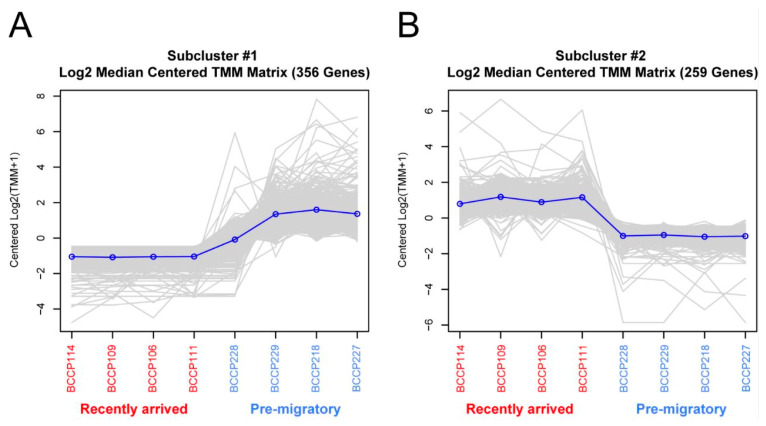
Differential gene expressions on subclusters 1 (**A**) and 2 (**B**) indicated in Figure 2. The expression in recently arrived (red) and pre-migratory (blue) individuals is expressed in *y* axis as TMM (Trimmed Mean of M-values; M = log_2_RA/PM); for detailed explanation, see [42].

**Table 1 ijms-24-12712-t001:** *Calidris pusilla* top 11 differentially expressed genes and their respective enriched Biological Process GO terms showing significant differential expressions in the brains of autumn recently arrived and spring pre-migratory semipalmated sandpipers.

Sequence ID	Gene Symbol	GO IDs	GO Names
Up-Regulated Genes			
TRINITY_DN40544_c0_g1	*GABBR2*	GO:0007186; GO:0007214;	G protein-coupled receptor signaling pathway; Gamma-aminobutyric acid signaling pathway;
TRINITY_DN3467_c1_g1	*MAN1A2*	GO:0005975; GO:0006491;	Carbohydrate metabolic process; N-glycan processing;
TRINITY_DN41739_c0_g1	*BCAT1*	GO:0009082; GO:0009098; GO:0009099;	Branched-chain amino acid biosynthetic process; Leucine biosynthetic process; Valine biosynthetic process;
TRINITY_DN56325_c0_g1	*NXPE3*	GO:0008150;	Encode a member of neurexophilin family of neuropeptide-like glycoproteins
TRINITY_DN62709_c0_g1	*FGF9*	GO:0000122; GO:0001525; GO:0001649; GO:0001654; GO:0001934; GO:0002053; GO:0002062; GO:0006606; GO:0008543; GO:0008584; GO:0010628; GO:0030178; GO:0030238; GO:0030326; GO:0030334; GO:0030949; GO:0032927; GO:0042472; GO:0045880; GO:0048505; GO:0048566; GO:0048706; GO:0050679; GO:0051781; GO:0060045; GO:0060484; GO:0090263; GO:1904707;	Negative regulation of transcription by RNA polymerase II; Angiogenesis; Osteoblast differentiation; Eye development; Positive regulation of protein phosphorylation; Positive regulation of mesenchymal cell proliferation; Chondrocyte differentiation; Protein import into nucleus; Fibroblast growth factor receptor signaling pathway; Male gonad development; Positive regulation of gene expression; Negative regulation of Wnt signaling pathway; Male sex determination; Embryonic limb morphogenesis; Regulation of cell migration; Positive regulation of vascular endothelial growth factor receptor signaling pathway; Positive regulation of activin receptor signaling pathway; Inner ear morphogenesis; positive regulation of smoothened signaling pathway; Regulation of timing of cell differentiation; Embryonic digestive tract development; Embryonic skeletal system development; Positive regulation of epithelial cell proliferation; Positive regulation of cell division; Positive regulation of cardiac muscle cell proliferation; Lung-associated mesenchyme development; Positive regulation of canonical Wnt signaling pathway; Positive regulation of vascular associated smooth muscle cell proliferation;
TRINITY_DN147173_c0_g1	*TRDMT1*	GO:0030488; GO:0036416;	tRNA methylation; A stabilization;
**Downregulated Genes**			
TRINITY_DN1477_c0_g1	*PLXNA1*	GO:0007162; GO:0008360; GO:0030334; GO:0043087; GO:0050772; GO:1902287	Negative regulation of cell adhesion; Regulation of cell shape; Regulation of cell migration; Regulation of GTPase activity; Positive regulation of axonogenesis; Semaphorin-plexin signaling pathway involved in axon guidance;
TRINITY_DN161848_c0_g1	*BICRAL*	GO:0045893;	Positive regulation of DNA-templated transcription;
TRINITY_DN51404_c0_g1	*ARHGEF9*	GO:0050790;	Regulation of catalytic activity;
TRINITY_DN123969_c0_g1	*MVB12B*	GO:0015031; GO:0019075; GO:0042058; GO:0046755;	Protein transport; Virus maturation; Regulation of epidermal growth factor receptor signaling pathway; Viral budding;
TRINITY_DN13053_c4_g1	*MAN1C1*	GO:0005975; GO:0006491; GO:0004571; GO:0005509; GO:0000139; GO:0005783; GO:0016020	Carbohydrate metabolic process; *N*-glycan processing; Mannosyl-oligosaccharide 1,2-α-mannosidase activity; Calcium ion binding; Golgi membrane; Endoplasmic reticulum; membrane

## Data Availability

The authors declare that, under request, all qualitative and quantitative data will be shared and genetic information, such as raw RNA-Seq data, has be deposited in the NCBI SRA at the following link https://www.ncbi.nlm.nih.gov/bioproject/PRJNA898584 (accessed on 30 June 2023).

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
