# Peer review of "Molecular Changes in the Brain of the Wintering *Calidris pusilla* in the Mangroves of the Amazon River Estuary"

_ijms, 2023, doi:10.3390/ijms241612712_

Round 1
Reviewer 1 Report
The authors investigate expression differences in semipalmated sandpiper after a long-distance migration to the wintering ground (recently arrived) and prior to the start of the migratory event back to the breeding ground (premigratory). Ultimately, they want to determine whether the genes they found are consistent with the described cellular changes they had previously investigated. While they did find some interesting results, I found the paper difficult to read, in part due to the organization of the results and discussion.
Major concerns
1. The discussion of the genes was super disconnected and needs some real work. The authors first focus on GABBR2 and ARHGEF9, but the ultimate link between GABBR2 and ARHGEF9 is not made until much later in the paragraph/section. Do they work in opposition of each other? Also, they link GABBR2 to GABAergic transmission but not how that links to hippocampal circuits and memory and such. Actually, the authors provide said link on line 263-265, not at line 247 where authors first bring up GABAergic activity. There needs to be a real overhaul of this section, with clear transitions for what the authors want to convey about the genes.
2. While the authors find some interesting expression differences between RA and PM birds, that clearly links to previous cellular differences, the narrative of their discussion does not flow. I’d like to see the authors clearly state what they found, and then explore what this means based on studies they had done previously, or what has been found in other species. In some of the discussion, a paragraph will start with what others found, and end with what the authors discovered. This organization does not work for the flow of the manuscript.
3. The description of the volcano plot seems excessive in the results. Might be better placed in the material and methods.
Minor comments
Italicize Calidris pusilla throughout manuscript (not italicized in abstract)
Line 34: de novo
Line 35: duplicated neurotranscriptome: rather say ‘compared expression levels between RA and PM birds’
Line 43-46 in abstract: Seems duplicated from what was previously stated above, but with less detail.
Lines 61-62: wording is off: should be leading to distinct regulatory strategies at the transcriptional level in autumn and spring.”
Lines 63: remove parentheses around (Emberiza melanocephala) and just bracket by commas
Lines 63: photoperiod instead of photoperiodical
Lines 63-64: wording is weird: change “it has been described, contrasting” to “researchers discovered contrasting…”
Line 67-69: sentence is out of place
Line 76-79: another rogue sentence. Does not need to be “paragraph” of own. Note, wording is confusing here. Do you mean combination instead of association?
Line 105: Not a new paragraph, add to line 104
Line 113: specify again that the 8 samples, 4 were RA and 4 were PM: Notes, this sample size seems small for DE study comparison- was it not possible to sample 10 birds per time-period?
Line 121: P-values should be p-values unless at start of sentence- no need to capitalize the P
Line 117-127: The author spend a lot of time explaining a volcano plot. The could just cite 41, and maybe a sentence about the interpretation of p-values in a volcano result, but should get right to the results. Authors do not need this entire section.
Line 154: To create a large-scale snap shot and clustering of differential expression in the brain of RA and PM C. pusilla data sets, we created a heatmap XX. We identify a unique differential expression profile for each cluster among the experimental groups. Why did you create heat map, and use active voice to say what you found…
Line 183: Why use recently arrived and pre-migratory when you’ve already set up the RA vs PM
Line 199: the fact that these genes are involved in various biological processes is super vague. Split it up and talk about which genes are involved in signaling pathways, which ones are involved in carbohydrate metabolism, etc
Line 203-208: run on sentence and a mix of too many details and too few. Additional genes: how many genes, what are they? Do I care about astrocyte differentiation and if I do, why (this seems like too much detail). Why name the top 20 genes and then have this vague sentence about other differential genes?
Line 221: remove comma after gene
Line 242, Begin sentence with “We identified significant differences…” and just cite Figure 1
Line 243-244: I don’t understand setting up sentence this way: Instead of:
Here we discuss GABBR2 and ARHGEF9 gene expressions as an example of two contrasting differential gene expressions related to inhibitory activity in the brain of C. pusilla
to “Notably, GABBR2 and ARHGEF9 exhibit contrasting differential gene expressions related to inhibitory activity in the brain of C. pusilla”
Line 309: define what astrocyte metabolism is: what is an astrocyte? How does it relate to lipid reserves? Transitions here are key too!
Lines 316-323: astrocyte-dopaminergic neuron signaling (CTNNB1) was down regulated but another one was unregulated (ATXN1). Both had to do with astrocyte- but their actual function wasn’t explored. Does it make sense that one was upregulated and another down regulated? More exploration/analysis is needed besides just naming genes.
Line 324: This paragraph sets up the DE of the immune response. The authors should first introduce DE of immune response that they found (APP), and then how it could relate to PM birds. First based on what they found (DE of several virus genomes), and then the viral genomes in other bird orders. The way it is written now, the reference 69 sounds like something the authors did in this paper, not what was previously found. This comment revolves around a major comment I have about the organization of the discussion.
Line 351: again, start with the fact you found NLRP3 was differentially expressed in this study, and then how it is important in birds. That would work for much of the discussion, the organization should be what the authors found and how it is supported by previous research in other systems.
Line 414: cite your work on the neurogenesis changes you found previously
Table 1 is the top 20 differentially expressed genes, with GO terms associated with them. It is Not the top 20 GO terms identified. Table legend is off, and again, the unknown genes are not necessary in this table. Also need to fix this in text 188-189
Figure 1: I like that the authors highlighted the known genes here, but think added the unknown genes makes the figure too busy
I believe this article requires moderate editing of English language. Some of this was due to jumbled sentences that were hard to follow. More so, it was the transitions between sentences and within paragraphs.
Reviewer 2 Report
I appreciate the author presenting this research article. My comments are as follows
1. Abstract is too long
2. How to decide which animal to catch? What is the basic information about their such as age, sex, weight and health condition?
3. The results only confirmed that the performance of the samples obtained at the two different time periods was different. However, the authors have tried to explain the results of the analyses, although the inference is reasonable, but there is no actual data to support them e.g. no information on Type I Interferons, IL-6, IL1-β, TNF and NF-kB, anti-inflammatory cytokines (IL-10), etc. is presented. It seem too over.
